# Assessment of NK Cell Activity Based on NK Cell-Specific Receptor Synergy in Peripheral Blood Mononuclear Cells and Whole Blood

**DOI:** 10.3390/ijms21218112

**Published:** 2020-10-30

**Authors:** Jung Min Kim, Eunbi Yi, Hyungwoo Cho, Woo Seon Choi, Dae-Hyun Ko, Dok Hyun Yoon, Sang-Hyun Hwang, Hun Sik Kim

**Affiliations:** 1Asan Medical Center, Department of Biomedical Sciences, University of Ulsan College of Medicine, Seoul 05505, Korea; kjm519999@naver.com (J.M.K.); rainyis0603@gmail.com (E.Y.); doloref@naver.com (W.S.C.); 2Asan Medical Center, Department of Oncology, University of Ulsan College of Medicine, Seoul 05505, Korea; hwcho@amc.seoul.kr (H.C.); dhyoon@amc.seoul.kr (D.H.Y.); 3Asan Medical Center, Department of Laboratory Medicine, University of Ulsan College of Medicine, Seoul 05505, Korea; daehyuni1118@amc.seoul.kr; 4Stem Cell Immunomodulation Research Center, University of Ulsan College of Medicine, Seoul 05505, Korea

**Keywords:** natural killer cells, NKG2D, 2B4, activity assay, multiple myeloma, TGF-β1

## Abstract

Natural killer (NK) cells are cytotoxic innate lymphocytes endowed with a unique ability to kill a broad spectrum of cancer and virus-infected cells. Given their key contribution to diverse diseases, the measurement of NK cell activity (NKA) has been used to estimate disease prognosis or the effect of therapeutic treatment. Currently, NKA assays are primarily based on cumbersome procedures related to careful labeling and handling of target cells and/or NK cells, and they require a rapid isolation of peripheral blood mononuclear cells (PBMCs) which often necessitates a large amount of blood. Here, we developed an ELISA-based whole blood (WB) NKA assay involving engineered target cells (P815-ULBP1+CD48) providing defined and synergistic stimulation for NK cells via NKG2D and 2B4. WB collected from healthy donors (HDs) and patients with multiple myeloma (MM) was stimulated with P815-ULBP1+CD48 cells combined with IL-2. Thereafter, it utilized the serum concentrations of granzyme B and IFN-γ originating in NK cells as independent and complementary indicators of NKA. This WB NKA assay demonstrated that MM patients exhibit a significantly lower NKA than HDs following stimulation with P815-ULBP1+CD48 cells and had a good correlation with the commonly used flow cytometry-based PBMC NKA assay. Moreover, the use of P815-ULBP1+CD48 cells in relation to assessing the levels of NKG2D and 2B4 receptors on NK cells facilitated the mechanistic study and led to the identification of TGF-β1 as a potential mediator of compromised NKA in MM. Thus, our proposed WB NKA assay facilitates the reliable measurement of NKA and holds promise for further development as both a clinical and research tool.

## 1. Introduction

Natural killer (NK) cells are a subset of innate lymphoid cells that constitute 5–15% of mononuclear cells in the peripheral blood [1,2]. NK cells are key effectors against a broad spectrum of tumor cells, virus-infected cells, or certain intracellular pathogens via granzyme/perforin-mediated direct cytolysis and production of inflammatory cytokines such as IFN-γ and TNF-α [3,4]. With an array of activating and inhibitory receptors, NK cells recognize and rapidly respond to their potential target cells and can trigger their effector functions without prior stimulation [1,5]. A distinct feature of NK cell activation is that these cells require the engagement of multiple activating receptors, which are not activating on their own but can cooperate in synergy, to elicit effective natural cytotoxicity against target cells [6]. One representative example of synergizing receptors for the activation of freshly isolated NK cells is NKG2D (CD314) paired with 2B4 (CD244) [7,8]. Cognate ligands for NKG2D include unconventional MHC class I-like molecules such as UL16-binding protein (ULBP) 1–6 and MICA/B, which are upregulated on stressed, virus-infected, or transformed cells [9]. The 2B4 receptor primarily recognizes CD48 on hematopoietic target cells and thus plays a key role in the surveillance of hematologic malignancies [10].

It has been established that NK cell functional deficiency is linked to an increased risk of developing diverse cancer types [11,12]. In support, many studies have revealed that NK cell effector functions are severely impaired in patients with different cancer types, correlating with a poor prognosis and increased mortality [13,14,15]. Consistently, the highly cytotoxic activity of circulating NK cells and/or significant tumor infiltration by these cells represents a positive prognostic indicator in various cancers [16,17]. Moreover, patients with an impaired function or development of NK cells suffer from recurrent illness associated with viral infections despite having functional adaptive immunity [18,19]. Thus, NK cell effector function can be regarded as a surrogate marker of intrinsic functional immunity against transformation and viral infection. In this regard, the measurement of NK cell activity (NKA) can be utilized as a prognostic biomarker, particularly in patients for monitoring disease progression or the effect of therapeutic treatment [11,20,21].

A number of different assays have been developed to assess functional NKA. The standard method is a cytotoxicity assay to evaluate the killing capacity of NK cells against tumor cells labeled with radioactive material (^51^Cr) or fluorescent probes (Europium or Calcein) [7,22,23]. These assays are based on the measurement of free radioactivity or fluorescence in the supernatant after the lysis of target cells by the effector cells. However, these methods are cumbersome and have technical shortcomings including poor labeling, high spontaneous release of labeled marker from some target cells, and the need to handle radioactive material. To overcome these difficulties, a flow cytometry (FC)-based assay for measuring NKA was developed. Cytotoxicity is determined with this method by the measurement of CD107a surface expression on the NK cells, and cytokine production was performed via the intracellular staining of the cytokines of interest in NK cells [24,25,26]. FC-based assays also enable the simultaneous monitoring of the effector functions of different NK cell subsets, based on the relative expression of surface markers (e.g., CD16 and CD56). However, these assays usually require a relatively large amount of whole blood (WB) and involve a separation step of peripheral blood mononuclear cells (PBMCs) or NK cells. Hence, there is an increasing demand for the development of NKA assessment methods that are convenient and reliable, preferably with a small volume of WB. Moreover, NKA assays using WB better reflect the physiological conditions under which NK cells operate [27].

Another important consideration for NKA measurement is the stimulation of NK cells using relevant and sensitive target cells. Human leukocyte antigen (HLA)-null cells such as myelogenous leukemia K562 and B-lymphoblastoid 721.221 cells are conventional target cells to stimulate NK cells through multiple ligand-receptor interactions [28,29,30]. These cells express multiple but as yet unidentified ligands for NK cell-activating receptors including NKG2D, DNAM-1, NKp30, NKp44, 2B4, and LFA-1. Given a degree of variations in the expression of activating receptors in different disease entities and between healthy individuals [31], NK cell encountering with these target cells may lead to heterogeneous receptor-ligand interactions and thereby complicates the accurate measurement of NKA [32]. In this regard, we previously introduced unrelated mouse target cells engineered to express defined ligands for human NKG2D and 2B4 (P815-ULBP1+CD48) [8,21]. These engineered target cells provide selective, uniform, and synergistic stimulation of NK cells by engaging NKG2D and 2B4 receptors that exhibit high-level constitutive expression on resting NK cells. Coexpression of ULBP1 and CD48 on insect S2 target cells also synergizes for cytotoxic degranulation of resting NK cells [33]. Compared with K562 target cells, NKA measurements with our engineered mouse target cells can reveal greater functional differences between healthy controls and patients with pancreatic cancer (PC) and thus have greater prognostic value in terms of recurrence and survival outcomes [21]. Although specific to NK cells, this assay was performed with PBMCs but has yet to be used to evaluate NKA using WB samples in comparison to PBMC samples.

In this study, using our engineered target cells that trigger NK cell-specific receptor synergy, we developed a NKA assay that assesses both NK cell cytotoxic activity and IFN-γ production in WB samples. We then compared the findings using this new NKA assay with those obtained from an established FC-based NKA assay employing PBMCs. To probe the potential clinical value of this assay, we used it to assess the NKA of patients with multiple myeloma (MM) in comparison to healthy individuals and compared these data with the results obtained with conventional K562 and 721.221 cells.

## 2. Results

### 2.1. Comparison of FC-Based NKA Using PBMCs from Healthy Donors and MM Patients

First, we used three different target cells for NKA assay to probe NK cell effector functions depending on differential stimulation. These were conventional HLA-null K562, 721.221, or unrelated mouse P815 cells engineered to express human ULBP1 and CD48, providing a defined and uniform stimulation. As the FC-based PBMC NKA assay is established and useful for measuring NKA on a per cell basis [11,24], we conducted CD107a degranulation and IFN-γ intracellular cytokine staining (ICS) assays using PBMC samples. In addition, NKA from patients with MM were also used and compared with those of healthy donors (HDs) to investigate the potential value for the clinical utility. Consistent with previous reports [8,21], P815-ULBP1+CD48 target cells were confirmed to trigger cytotoxic degranulation and IFN-γ production in a NK cell-specific and synergistic manner, whereas stimulation with ULBP1 or CD48 alone induced little NK cell activation (Appendix A). Incubation with all three target cells for 2 h resulted in a significant increase in CD107a expression on NK cells in PBMC samples from HDs but in the order of their activation capacities: P815-ULBP1+CD48 > 721.221 > K562 (Figure 1A,B). Moreover, NK cells from the MM group exhibited a significant impairment in cytotoxic degranulation compared with those from HDs in response to P815-ULBP1+CD48 and 721.221 cells but not K562 cells (*p* < 0.01 against P815-ULBP1+CD48; *p* < 0.05 against 721.221) (Figure 1A,B). Next, we assessed the capacity of NK cells to produce IFN-γ. A significant increase in IFN-γ-positive NK cells was observed in HD PBMC samples following stimulation with all three target cells for 6 h with 721.221 and P815-ULBP1+CD48 cells showing comparable and most potent effects (Figure 1C,D). As observed with NK cell degranulation, NK cells from the MM group also produced a significantly less IFN-γ than those from HDs against P815-ULBP1+CD48 and 721.221 cells but not K562 cells (*p* < 0.01 against P815-ULBP1+CD48; *p* < 0.05 against 721.221) (Figure 1C,D). However, the frequencies of NK cells are comparable between the HD and MM groups (Appendix A). Thus, by assessing the NKA of individual cells, NK cells of MM group were clearly impaired in their ability to trigger cytotoxic degranulation and IFN-γ production, which was most pronounced upon the stimulation with P815-ULBP1+CD48 target cells.

### 2.2. Comparison of ELISA-Based NKA Using WB Samples from Healthy Donors and MM Patients

Despite the observations of the usefulness of P815-ULBP1+CD48 target cells in terms of NK cell specificity and clinical applicability, a simpler procedure for the NKA assay is desirable in clinical practice. Thus, we developed an ELISA-based NKA assay using WB without the need to isolate PBMC or NK cells (Figure 2A). WB samples collected from HDs and MM patients were coincubated with K562, 721.221, or P815-ULBP1+CD48 target cells for 24 h in the presence of 100 U/mL IL-2 as the activation with IL-2 amplified the NK cell response without hampering the performance of the assay. After the stimulation, the concentrations of granzyme B and IFN-γ in the supernatant of incubation mixture were determined by ELISA. The results revealed that granzyme B release was significantly lower in the MM group compared to the HD group in response to P815-ULBP1+CD48 and K562 cells but not 721.221 cells (*p* < 0.001 against P815-ULBP1+CD48; *p* < 0.05 against K562) (Figure 2B). Similarly, the levels of IFN-γ were significantly lower in the MM group than in the HD group in response to all three target cell types with the most significant effect by P815-ULBP1+CD48 target cells (*p* < 0.001 against P815-ULBP1+CD48; *p* < 0.01 against K562; *p* < 0.05 against 721.221).

Next, we compared the results of ELISA-based WB NKA assay with those of the established FC-based NKA using PBMCs. As granzyme B is a cytotoxic effector molecule that is exocytozed toward target cells during NK cell degranulation, we analyzed the correlation between the results of the WB granzyme B release assay and the PBMC CD107a degranulation assay. We observed a significant positive correlation between the levels of granzyme B release and the percentages of CD107a-postitive NK cells following stimulation with all three target cells (*p* < 0.01 against P815-ULBP1+CD48; *p* < 0.05 against K562; *p* < 0.05 against 721.221) (Figure 3A), highlighting the validity of our WB NKA assay. In addition, the levels of IFN-γ secretion in WB positively correlated with the percentages of IFN-γ-positive NK cells in PBMCs in response to P815-ULBP1+CD48 cells (*p* < 0.05) but not K562 and 721.221 cells (Figure 3B). Based on these results, we conclude that our ELISA-based WB NKA assay is a viable alternative to the commonly used FC-based PBMC NKA assay, particularly if using P815-ULBP1+CD48 target cell stimulation.

### 2.3. NKG2D Expression Being Associated with NK Cell Degranulation in Healthy Donors and MM Patients

NK cell activation is triggered by diverse activating receptors that are often downregulated as a consequence of NK cell dysfunction in various cancers [11,34]. Among other receptors, a decrease in NKG2D and/or 2B4 has been implicated in the NK cell defects seen in MM [35,36]. Moreover, as P815-ULBP1+CD48 target cells activated NK cells through NKG2D and 2B4, we assessed their expression on NK cells in MM patients in comparison to HDs. There were no significant differences in the expression of NKG2D or 2B4 between the two groups, despite a clear decrease of NKG2D expression in some MM patients (Figure 4A,B). Given the involvement of NKG2D and 2B4 receptors in MM, we assessed the correlation between these receptors and NK cell functions to probe their potential contribution to NK cell dysfunction. The expression of 2B4 did not correlate with the frequencies of CD107a- or IFN-γ-positive NK cells from the two groups in response to any of the three target cells (Figure 4C). By comparison, NKG2D expression positively correlated with the frequency of CD107a-positive but not IFN-γ-positive, NK cells following stimulation with P815-ULBP1+CD48 and K562 cells (*p* < 0.01 against P815-ULBP1+CD48; *p* < 0.05 against K562) (Figure 4D). The lack of association of NK cell function with the 2B4 level and the weak association with the NKG2D level, as evident from P815-ULBP1+CD48 cell stimulation, suggested that mechanisms other than receptor downregulation could contribute to the compromised function of NK cells in MM.

### 2.4. TGF-β1 Levels Being Related to Impaired NKA in MM

Next, we assessed the levels of soluble mediators implicated in NK cell dysfunction in the plasma of the MM patients and HDs. Compared with the HD group, the MM group had significantly higher levels of TGF-β1, a potent immunosuppressor of NK cell functions [37,38] that is actively secreted by the malignant plasma cells in MM (Figure 5A). By comparison, the levels of PGE_2_ were below the detection limit in both groups (data not shown). Of note, the levels of TGF-β1 correlated inversely with the frequencies of CD107a- and IFN-γ-positive NK cells after stimulation with P815-ULBP1+CD48 cells but not with the other target cells (CD107a+: *p* < 0.05; IFN-γ+: *p* < 0.001) (Figure 5B). In addition, TGF-β1 levels were inversely correlated with the levels of granzyme B and IFN-γ release in WB following stimulation with P815-ULBP1+CD48 cells (granzyme B: *p* < 0.01; IFN-γ: *p* < 0.001) and K562 cells (IFN-γ: *p* < 0.01) (Figure 5C), implying TGF-β1 as a potential mediator of compromised NKA in MM. Thus, our data suggest that the assessment of NKA via stimulation with defined receptor-ligand interactions, herein NKG2D and 2B4, could be utilized as a supportive diagnostic tool, or as an experimental system for further elucidating the mechanism of NK cell dysfunction in MM.

## 3. Discussion

NK cells serve a first line of defense against a broad spectrum of cancer cells and virus-infected cells without the restriction to major histocompatibility complex (MHC) specificity. Given the close association between NK cell functions and the outcome of diseases including cancer, the assessment of circulating NK cell function is considered a useful measure for monitoring disease progression or the effect of therapeutic treatment. In this study, we developed an ELISA-based WB NKA assay involving engineered target cells providing defined and uniform stimulation for NK cells via NKG2D and 2B4. In addition to the requirement for a small volume of WB (500 μL), this methodology has proven to be convenient and to show good correlation with the commonly used FC-based PBMC NKA assay. Moreover, we demonstrated here the potential value and applicability of this assay in a clinical setting by assessing the NKA in MM patients in comparison to HDs. In these analyses, the MM patients exhibited a significantly lower NK cell production of granzyme B and IFN-γ than the HD subjects, which was a more prominent difference when using P815-ULBP1+CD48 target cells compared with the other target cells. Finally, the use of P815-ULBP1+CD48 cells enabled us to evaluate whether the compromised NKA in MM NK cells is related to NKG2D and/or 2B4 downregulation by assessing the expression of these receptors on NK cells. This simplified our subsequent mechanistic analyses by allowing us to focus on defined receptors, and facilitated the identification of TGF-β1 as a potential mediator of NK cell dysfunction in MM. These findings together highlight the feasibility and utility of NKA measurement via stimulation with defined receptor-ligand interactions, hereby ULBP1 for NKG2D and CD48 for 2B4.

A WB-based NKA assay has several advantages over a PBMC-based NKA assay. It is a simpler and more convenient approach without the need to purify PBMCs, which is often expensive and both time- and labor-consuming due to the several isolation steps. In addition, a WB-based assay is suitable for monitoring NKA in smaller amounts of blood, which is very useful for certain patients, particularly pediatric patients or patients with lymphocytopenia. Moreover, a WB NKA assay may better reflect the in vivo state of NK cells by providing physiological conditions for NK cell activation due to the preservation of soluble factors (such as cytokines) and of interaction with other immune cells that cooperatively affect NK cells. Although several WB NKA assays have been developed [22,39,40], there are a few caveats regarding their application to frequent monitoring of NKA in clinical practice. These assays are performed with target cells labeled with hazardous radioactive material or fluorescent probes and need to be carefully optimized for conditions regarding the labeling, incubation, and post-handling of labeled target cells. Alternatively, some of the assays required several fluorochrome-conjugated antibodies and many experimental steps for measuring CD107a degranulation and intracellular IFN-γ expression in NK cells by flow cytometry. Moreover, the performance and reproducibility of these assays is limited by their low sensitivity when compared with PBMC-based assay, which is likely due to the relatively low number and activation levels of NK cells in WB. To increase this sensitivity, it was suggested to add optimal doses of cytokines involved in NK cell activation such as IL-2, IL-15, or IL-12/IL-18 during the WB NKA assay [22]. In this study, after careful titration of the IL-2 dose, we developed a reproducible WB NKA assay in the presence of 100 U/mL IL-2 based on a simple ELISA procedure, i.e., mixing WB with target cells (e.g., P815-ULBP1+CD48) in IL-2 containing media and then analyzing the amount of granzyme B and IFN-γ in the supernatant. Moreover, our WB NKA assay showed a comparable sensitivity and correlated well with the commonly used FC-based PBMC NKA assay, thus validating its performance.

The serine protease granzyme B is a key cytotoxic effector molecule released from the granules of cytotoxic lymphocytes, and IFN-γ is the major activator of cell-mediated immunity through the orchestration of inflammatory responses [41,42]. Both are important markers of cytotoxic lymphocyte activity and are released upon incubation with sensitive target cells [8,43]. By comparison, NK cell activation through soluble factors (e.g., IL-12/IL-18) primarily stimulates the production of cytokines such as IFN-γ [44]. Similarly, an ELISA-based WB NKA assay was recently developed to measure the levels of IFN-γ release in WB after stimulation with Promoca, a combination of recombinant cytokines of unknown identity [45,46]. However, this assay is unable to assess the cytotoxic activity of NK cells and has an issue in relation to the specificity of IFN-γ production by NK cells. In this regard, the simultaneous measurement of NK cell cytolytic activity and IFN-γ production is desirable to gain precise information about NKA and disease progression. This notion is supported by numerous case studies [13,14,15] as well as a seminal prospective study that reported a high incidence of diverse cancers in subjects with low cytotoxic activity of peripheral NK cells [12]. We also demonstrated in our previous study that the cytotoxic degranulation, but not IFN-γ production, of peripheral NK cells correlates well with PC progression [21].

MM is a malignant disease of plasma cells and is characterized by immune dysregulation, including impaired NK cell functions [20,47]. It has been shown that the progression of MM is associated with NK cell cytotoxic activity, including a clinical stage-dependent decrease in NKA and in the number of bone marrow NK cells [20,48]. Consistent with this, we here observed impaired effector functions of NK cells in the MM group compared with the HD group using either the FC-based PBMC NKA assay or ELISA-based WB NKA assay, particularly with the P815-ULBP1+CD48 target cell stimulation. The downregulation of activating receptors is considered to be a major mechanism of NK cell dysfunction in various cancers, including MM [35,36]. A decrease in the expression of NKG2D and/or 2B4 was reported on NK cells from MM patients, which likely contributes to the escape of MM cells from NK cell recognition. However, we observed no significant downregulation of NKG2D and 2B4 in our cohort of MM patients and no strong correlation of NK cell function with these levels either, despite a weak association with NKG2D. These results suggest a potential involvement of other mechanisms underlying NK cell dysfunction. In support, we found significantly elevated levels of TGF-β1 in the plasma of MM patients and significant inverse correlation between the TGF-β1 levels and NKA. Malignant plasma cells and regulatory T cells in MM secrete high levels of TGF-β1, which contributes to the suppression of NK cell function [37,38]. Despite the significant therapeutic progresses, more effective strategies to combat this disease are required due to incurable relapsed and refractory MM [11]. In this regard, our data suggest that our new ELISA-based WB NKA assay, probably in combination with the measurement of TGF-β1 levels, has potential value and warrants further development as a supportive diagnostic or prognostic tool for MM in treatment-naïve or posttreatment situations. However, given a relatively small sample size enrolled in this study, further study is required to validate our results in a larger cohort of patients, along with sufficient amount of blood for mechanistic analyses.

In summary, we described here a convenient and reliable method of measuring NKA using a small volume of WB in comparison to the commonly used FC-based PBMC NKA assay. In addition, our WB NKA assay has potential clinical application for NKA assessments in MM patients and for subsequent mechanistic studies. Although further validation is required, our findings may also provide support for therapeutic strategies such as targeting TGF-β1 aimed at restoring NK cell-mediated immune surveillance in MM.

## 4. Materials and Methods

### 4.1. Study Design and Patients

Among the patients who were diagnosed with multiple myeloma between April 2013 and December 2019 at Asan Medical Center, South Korea, 10 patients who agreed to this study were included. Peripheral blood samples were collected, and PBMCs and plasma were harvested within 1 h and then cryopreserved until use as described [21]. The following data were extracted from the medical records and analyzed: age, gender, disease setting, international staging system (ISS), revised-ISS, hemoglobin, total white blood cell counts, serum β2 microglobulin level, serum LDH level, serum albumin level, serum M-protein level, urine M-protein level, myeloma subtype and isotype, serum IgG level. As a control, 16 healthy volunteers were included. Patient clinical and demographic characteristics are summarized in Appendix A. All subjects gave their informed consent for inclusion before they participated in the study. The study was conducted in accordance with the Declaration of Helsinki, and the protocol was approved by the Institutional Review Board (IRB) of Asan Medical Center (2016–0007, 6 January 2016).

### 4.2. Target Cell and Culture

Mouse mastocytoma P815 cells with stable surface expression of ULBP1 (a ligand for human NKG2D) and CD48 (a ligand for human 2B4) were used to measure the effector function of NK cells in the context of defined and uniform stimulation [21]. The expression of ULBP1 and CD48 on engineered P815 cells was confirmed by flow cytometry, as described [8]. K562, 721.221, and P815-ULBP1+CD48 cells were maintained in Iscove’s modified Dulbecco’s medium (IMDM; Hyclone, Logan, UT, USA) supplemented with 10% fetal bovine serum (FBS; Gibco BRL, Waltham, MA, USA) and 2 mM L-glutamine. The cells were confirmed to be free of mycoplasma contamination.

### 4.3. Antibodies

The following fluorochrome-conjugated monoclonal antibodies (mAbs) were used to evaluate NKA by flow cytometry: anti-CD3-PerCP (clone SK7, BD Bioscience, San Jose, CA, USA), anti-CD56-PE (clone NCAM16.2, BD Bioscience, San Jose, CA, USA), anti-CD107a/LAMP1-FITC (clone H4A3, BD Bioscience, San Jose, CA, USA), and anti-IFN-γ-FITC (clone 25723.11, BD Bioscience, San Jose, CA, USA). The following fluorochrome-conjugated mAbs were used for the analysis of NKG2D and 2B4 on NK cells: anti-CD3-PerCP (clone SK7, BD Bioscience, San Jose, CA, USA), anti-CD56-FITC (clone NCAM16.2, BD Bioscience, San Jose, CA, USA), anti-CD314/NKG2D-PE (clone 149810, R&D Systems, Minneapolis, MN, USA), anti-CD244/2B4-PE (clone C1.7, BD Bioscience), and mouse IgG1 isotype control (clone MOPC-21, BD Bioscience, San Jose, CA, USA).

### 4.4. NK Cell Cytotoxic Degranulation and Intracellular IFN-γ Staining Assay

PBMCs were separated from WB by density gradient using a lymphocyte separating medium (LSM; MP biomedicals, Illkirch, France), frozen in FBS containing 10% DMSO (Sigma-Aldrich, St. Louis, MO, USA), kept at −80 °C for 24 h, and then stored at −196 °C in liquid nitrogen. Cryopreserved PBMCs including paired MM patients and healthy donors were thawed rapidly at 37 °C in a single day and suspended in complete RPMI medium (RPMI-1640 medium supplemented with 10% FBS, 2 mM L-glutamine, 100 U/mL penicillin, and 100 µg/mL streptomycin in the presence of 50 U/mL of DNase I (Roche, Basel, Switzerland)). The cells were washed twice and then resuspended with complete RPMI medium at ≈4 × 10^6^ cells/mL, followed by overnight resting.

Cytotoxic degranulation of NK cells was assessed by CD107a expression on the cell surface, as previously described [33]. Briefly, PBMCs (1 × 10^5^ cells) were incubated with an equal number of K562, 721.221, or P815-ULBP1+CD48 cells for 2h at 37 °C. After centrifugation, the cell pellets were resuspended in FACS buffer (PBS with 1% FBS) and stained with anti-CD3-PerCP, anti-CD56-PE, and anti-CD107a/LAMP1-FITC for 35 min in the dark at 4 °C. The forward and side scatter (FSC and SSC) characteristics were used to identify lymphocytes, and the CD107a expression on CD3-CD56+ NK cells was analyzed by flow cytometry using FlowJo software (ver.10, Tree Star, Ashland, OR, USA).

Cytokine production of NK cells was determined by intracellular expression of IFN-γ, as described [8]. PBMCs (1 × 10^5^ cells) were stimulated with an equal number of the indicated target cells for 1 h at 37 °C. Thereafter, brefeldin A (GolgiPlug; BD Bioscience, San Jose, CA, USA) and monensin (GolgiStop; BD Bioscience, San Jose, CA, USA) were added, and the samples were incubated for an additional 5 h, for a total of 6 h. After the incubation, the cells were first stained with anti-CD3-PerCP and anti-CD56-PE for 35 min in the dark at 4 °C. After washing twice with a FACS buffer, the PBMCs were fixed and permeabilized with a BD Cytofix/Cytoperm solution (BD Bioscience, San Jose, CA, USA) for 20 min in the dark at 4 °C. The cells were then washed twice with BD Perm/Wash buffer (BD Bioscience, San Jose, CA, USA) and stained with anti-IFN-γ-FITC for overnight in the dark at 4 °C. Thereafter, the expression of IFN-γ in CD3-CD56+ NK cells was analyzed by flow cytometry.

### 4.5. Whole Blood (WB) ELISA Assay

Heparinized WB samples (500 μL) were transferred to 1.5 mL microcentrifuge tubes and diluted 2-fold with an IMDM medium supplemented with 10% FBS, 1 mM sodium pyruvate, and 1× MEM nonessential amino acid solution (Gibco BRL, Waltham, MA, USA) in the presence of 100 U/mL recombinant IL-2 (Roche, Basel, Switzerland). The diluted WB samples were then incubated with 1 × 10^6^ K562, 721.221, or P815-ULBP1+CD48 target cells for 24 h at 37 °C in a MACSmix tube rotator. Thereafter, supernatants of the incubation mixture were collected by centrifugation, and the levels of granzyme B (R&D Systems, Minneapolis, MN, USA) and IFN-γ (Thermo Scientific) therein were determined by using ELISA according to the manufacturer’s instructions.

### 4.6. Plasma Cytokine Analysis

After blood component fractionation by density gradient centrifugation, plasma fractions were aliquoted and stored at −80 °C until analysis. The levels of TGF-β1 were quantified using the DuoSet ELISA (R&D Systems) according to the manufacturer’s instructions. The levels of serum PGE_2_ and PGD_2_ were measured by LC-MS/MS analysis as previously described [49].

### 4.7. Statistical Analysis

All data were analyzed by using GraphPad Prism v. 5.00 (GraphPad Software Inc., San Diego, CA, USA). Two groups were compared using nonparametric Mann–Whitney *U*-tests and Fisher’s exact tests (two-tailed). Correlations between two parameters were evaluated using the nonparametric Spearman’s rank correlation test. *p* < 0.05 was considered significant and the degree of significance is presented as follows: * *p* < 0.05, ** *p* < 0.01, and *** *p* < 0.001.

## Figures and Tables

**Figure 1 ijms-21-08112-f001:**
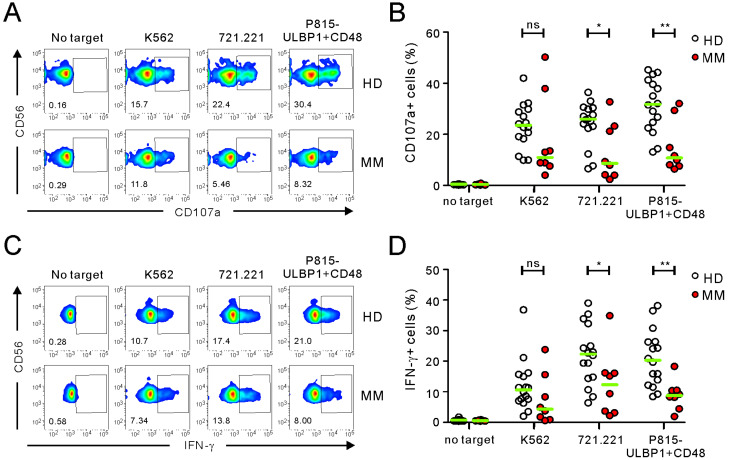
Comparison of natural killer (NK) cell functions between healthy donors (HDs) and patients with multiple myeloma (MM) using the PBMC-based NK cell activity (NKA) assay. Isolated peripheral blood mononuclear cells (PBMCs) from the healthy donor (HD) group (*n* = 16), and multiple myeloma (MM) patient group (*n* = 8) were co-cultured with K562, 721.221, or P815-ULBP1+CD48 target cells. (**A**,**B**) Cytotoxic degranulation of NK cells, as measured by cell surface expression of CD107a using flow cytometry. (**C**,**D**) IFN-γ production by NK cells, as measured by intracellular expression of IFN-γ using flow cytometry. Representative flow cytometry profiles (**A**,**C**) and summary graphs (**B**,**D**) showing the percentages of CD107a+ and IFN-γ+ NK cells. Horizontal bars (green) indicate the medians. Statistical differences between the groups were evaluated with the nonparametric Mann–Whitney *U*-test. * *p* < 0.05 and ** *p* < 0.01.

**Figure 2 ijms-21-08112-f002:**
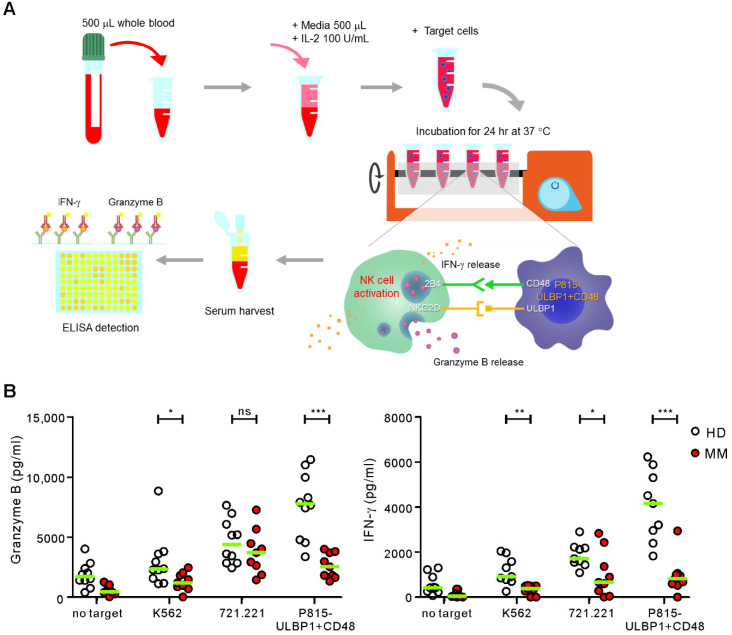
Comparison of NK cell functions between HDs and patients with MM using the whole blood (WB)-based NKA assay. WB samples from the HD group (granzyme B: *n* = 10; IFN-γ: *n* = 9) and MM group (*n* = 9) were incubated with the indicated target cells in the presence of 100 U/mL IL-2. (**A**) Scheme for NKA measurement using WB-NKA assay. (**B**) Secretion of granzyme B (**left**) and IFN-γ (**right**) into the supernatant was measured by ELISA. Horizontal bars (**green**) indicate the medians. * *p* < 0.05, ** *p* < 0.01, and *** *p* < 0.001; Mann-Whitney *U*-test.

**Figure 3 ijms-21-08112-f003:**
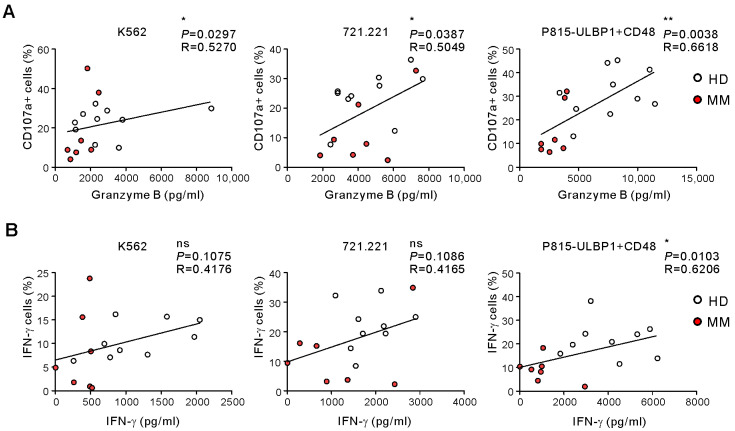
Correlation between PBMC-based NKA assay and WB-based NKA assay. (**A**) Correlation between granzyme B levels and the frequencies of CD107a+ NK cells. (**B**) Correlation between IFN-γ levels and the frequencies of IFN-γ+ NK cells. * *p* < 0.05 and ** *p* < 0.01; Spearman’s correlation test (**A**,**B**).

**Figure 4 ijms-21-08112-f004:**
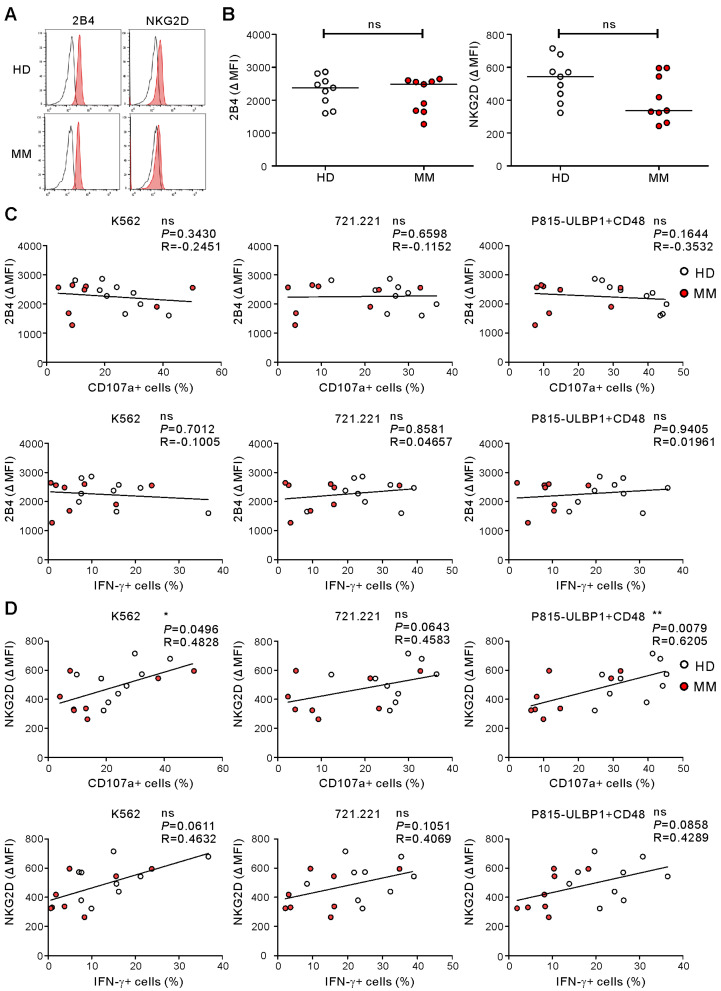
NKG2D expression is associated with NK cell degranulation in response to receptor coactivation. (**A**) Representative FACS profiles showing the expression of NKG2D and 2B4 on gated NK cells in the HD group and MM group. (**B**) Shown is the mean fluorescence intensity (MFI) of the expression of 2B4 and NKG2D on NK cells relative to the MFI of the isotype control (ΔMFI) in the HD group (*n* = 9) and MM group (*n* = 9). (**C**) Correlation between 2B4 expression (ΔMFI) and the frequencies of CD107a+ or IFN-γ+ NK cells. (**D**) Correlation between NKG2D expression (ΔMFI) and the frequencies of CD107a+ or IFN-γ+ NK cells. * *p* < 0.05 and ** *p* < 0.01; Spearman’s correlation test.

**Figure 5 ijms-21-08112-f005:**
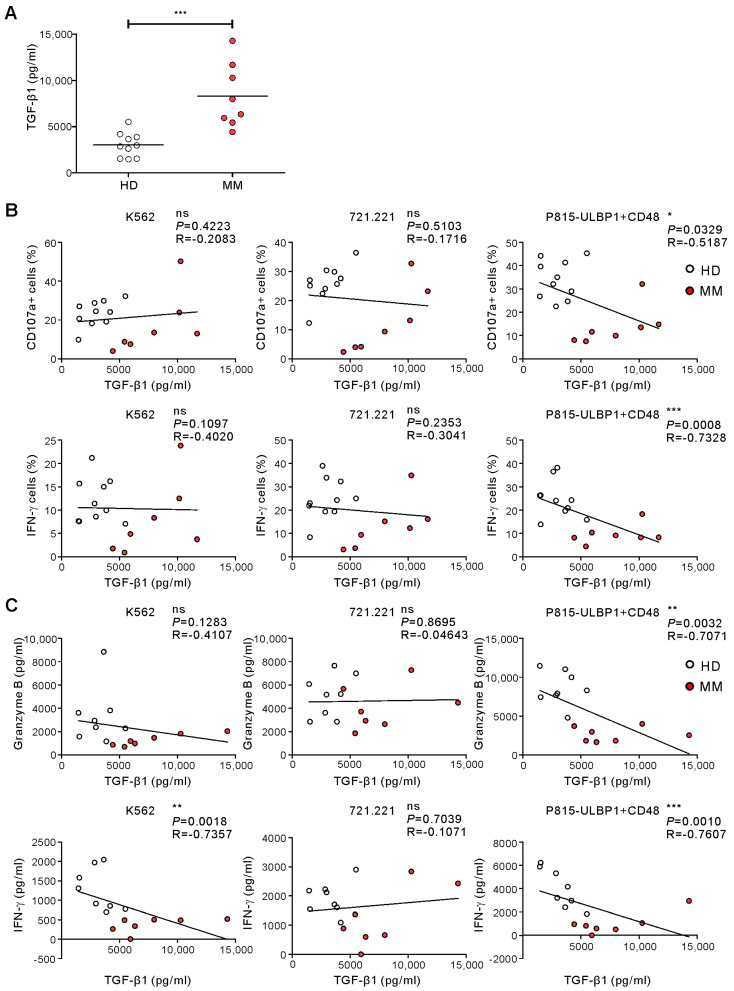
Correlation between the levels of TGF-β1 and impaired NKA in response to receptor coactivation. (**A**) The plasma levels of TGF-β1 was measured by ELISA in the HD group (*n* = 10) and MM group (*n* = 8). Horizontal bars denote the medians. (**B**) Correlation between plasma TGF-β1 levels and the percentages of CD107a+ or IFN-γ+ NK cells. (**C**) Correlation between plasma TGF-β1 levels and the levels of granzyme B or IFN-γ release. * *p* < 0.05, ** *p* < 0.01, and *** *p* < 0.001; Mann–Whitney U-test (**A**) and Spearman’s correlation test (**B**,**C**).

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
