# Peer review of "Assessment of NK Cell Activity Based on NK Cell-Specific Receptor Synergy in Peripheral Blood Mononuclear Cells and Whole Blood"

_ijms, 2020, doi:10.3390/ijms21218112_

Round 1
Reviewer 1 Report
The authors presents an interesting improvement of an already well established tecnique to easily evaluate NK cell activity in terms of IFN-gamma production and degranulation assay against NK susceptible target cells.This tecnique seems to display good performance either using isolated PBMC cell suspensions either whole blood samples.
This sytem may be useful in all situation ( such as analyses on large numrbes of patients) in which separation of PBMC may represent an issue , representIng an affordable system that allows rapid responses. However there are some concerns from a technical point of view and also from a the scientific point of view with regards to the data on MMM patients.
Major points
The analyses on the validation of their experimental system are well presented and quite appropriately performed. However, the choice of CD48 to support the stimulation of NKG2D by ULBP1 couldn't have the expected effects because 2B4 is extremely important in supporting the costimulation of CD16 or Natural Cytotoxicity Receptors (NCR), i.e. of itam motifs bearing receptors, while is less important in the costimulation of NKG2D or DNAM-1 ( Meazza et al. European Journal of Immunolgy 2014) receptors. For sure, every co-receptors may be useful to amplificate NK cell activation and the formation of an efficient immunological synapse, however it is possible that, in this situation, ULBP1 and CD48 do not work in a synergistic way. They could analyze the NK cell activity against P815 engineered for the expression of ULBP1 only, or for the expression of other ULBPs (2 or 3) which can be more easily detectable on several tumor targets.
There are some concerns on data and discussion on the reason why NK cell acitivity in MM patients is impaired. Beyond TGF-beta expression ( that may exert effects not only on NK receptors but also on adhesion molecules, Granzyme, Perforin expression )there is a lot of literature on this issue that they could exploit to better characterized the phenomena that they observe.
MInor points
Authors should show data obtained with wild type P815 cell line as negative control .
Author Response
Reviewer 1
The authors presents an interesting improvement of an already well established technique to easily evaluate NK cell activity in terms of IFN-gamma production and degranulation assay against NK susceptible target cells. This technique seems to display good performance either using isolated PBMC cell suspensions either whole blood samples.
This system may be useful in all situation (such as analyses on large numbers of patients) in which separation of PBMC may represent an issue, representing an affordable system that allows rapid responses. However there are some concerns from a technical point of view and also from the scientific point of view with regards to the data on MM patients.
Major points
The analyses on the validation of their experimental system are well presented and quite appropriately performed. However, the choice of CD48 to support the stimulation of NKG2D by ULBP1 couldn't have the expected effects because 2B4 is extremely important in supporting the costimulation of CD16 or Natural Cytotoxicity Receptors (NCR), i.e. of itam motifs bearing receptors, while is less important in the costimulation of NKG2D or DNAM-1 (Meazza et al. European Journal of Immunolgy 2014) receptors. For sure, every co-receptors may be useful to amplificate NK cell activation and the formation of an efficient immunological synapse, however it is possible that, in this situation, ULBP1 and CD48 do not work in a synergistic way. They could analyze the NK cell activity against P815 engineered for the expression of ULBP1 only, or for the expression of other ULBPs (2 or 3) which can be more easily detectable on several tumor targets.
Re) In normal NK cells, 2B4 has been shown to synergize with other activating receptors such as NKG2D, NKp46, and DNAM-1 for the activation of freshly isolated NK cells (Bryceson, Y.T. et al., Blood 2006, 107:159, PMID: 16150947). By comparison, in XLP1 NK cells lacking functional SAP, 2B4 impairs the activity of co-engaged activating receptors by delivering inhibitory signals, which is confined to ITAM-bearing receptors but not non-ITAM-associated NKG2D and DNAM-1 (Meazza, R. et al., Eur J Immunol. 2014, 44:1526, PMID: 24496997). In our study, ULBP1 (a ligand for human NKG2D) and CD48 (a ligand for human 2B4) were selected to determine the functional capacity of NK cells, on the basis of the confirmed findings that their coexpression on insect S2 target cells (Bryceson, Y.T. et al., Blood 2009, 114:2657, PMID: 19628705) as well as mouse P815 target cells (Kwon, H.J. et al., Nat Commun. 2016, 7:11686, PMID: 27221592; Jun, E. et al., Front Immunol. 2019, 10:1354, PMID: 31281312) synergizes for cytotoxic degranulation and IFN-g production of NK cells. As suggested by the Reviewer, in the revision, we incorporated the data analyzing the NK cell activity against wild type P815 cells or P815 cells engineered for the surface expression of ULBP1 or CD48 alone. Consistent with previous reports, receptor coactivation with P815 target cells expressing both ULBP1 and CD48 resulted in synergistic degranulation and IFN-g production in NK cell population, whereas stimulation with wild type P815 cells and P815 cells expressing ULBP1 or CD48 alone induced little activation of NK cells (Revised Supplementary Figure 1). Thus, we show evidence that P815-ULBP1+CD48 target cells provide selective, uniform, and synergistic stimulation of NK cells as a useful tool to measure NK cell function. In this regard, the text has been revised accordingly with additional reference (Line 48 for an additional reference; Lines 94~95; Lines 117~118 for Revised Supplementary Figure 1).
Reference added:
- Bryceson, Y.T.; March, M.E.; Ljunggren, H.G.; Long, E.O. Synergy among receptors on resting NK cells for the activation of natural cytotoxicity and cytokine secretion. Blood 2006, 107, 159-166.
There are some concerns on data and discussion on the reason why NK cell activity in MM patients is impaired. Beyond TGF-beta expression (that may exert effects not only on NK receptors but also on adhesion molecules, Granzyme, Perforin expression) there is a lot of literature on this issue that they could exploit to better characterize the phenomena that they observe.
Re) We agree with the Reviewer’s opinion that an impaired NK cell function observed in MM patients could be attributed to other mediator(s) or mechanism(s) beyond a significant association of TGF-b1, a potent suppressor of NK cell functions, in our study. Although the focus of our study is to investigate the feasibility and usefulness of P815-ULBP1+CD48 target cells for assessing functional capacity of NK cells in PBMC and whole blood samples, we performed a further study to highlight their utility in mechanistic analyses of NK cell dysfunction in MM patients. The use of P815-ULBP1+CD48 target cells enabled us to focus on defined receptors (i.e. NKG2D and 2B4) and facilitated the identification of TGF-b1 for correlating NK cell defect compared with other conventional target cells (i.e. K562) that express multiple and unidentified ligands for NK cell activating receptors. However, as commented by the Reviewer, we agree with the necessity for further mechanistic study with the inclusion of sufficient number of cases and amount of blood to establish mechanism(s) underlying NK cell dysfunction in MM patients. Unfortunately, besides limited time given for revision, comprehensive mechanistic study of NK cells from MM patients are not possible under our IRB approval that was expired recently and limited the amount of blood primarily to assessing the NK cell activity. Accordingly, we have toned down the importance of TGF-b1 in mediating NK cell dysfunction in MM patients and have discussed the requirement for further mechanistic study with the inclusion of sufficient number of cases and amount of blood in the revised manuscript (Line 34; Line 201; Lines 220~221; Line 249; Line 303; Lines 310~312).
Minor points
Authors should show data obtained with wild type P815 cell line as negative control.
Re) As suggested by the Reviewer, we included the data analyzing the NK cell activity against wild type P815 cell line, P815-ULBP1, or P815-CD48 cells in addition to P815-ULBP1+CD48 cells (Revised Supplementary Figure 1). This data support the synergistic activation of NK cells following stimulation with P815-ULBP1+CD48 cells (Lines 117~118 for Revised Supplementary Figure 1).

Reviewer 2 Report
The authors proposed a new target cell, P815-ULBP1+CD48, as a method of measuring new NKA (NK cell activity) and tried to prove its usefulness. The NK cytotoxicity of cells isolated from PBMC or WB of normal donor (HD) and MM (multiple myeloma) patients was compared with CD107a+ cell percentage, IFN-gamma+ cell percentage, and the amount of granzyme B and IFN-gamma secreted. There was a significant difference when P815-ULBP1+CD48 was used as the target cell. When P815-ULBP1+CD48 was used as the target cell, it was more significant than when the K562 cell, which is usually used, was used as the target. The authors argue that the use of P815-ULBP1+CD48 as a target cell is advantageous for predicting treatment in patients with MM.
Analysis of LAMC cancer in the TCGA data base (by me) showed that the expression levels of ULBP1 or CD48 did not significantly affect overall survival. However, the expression level of ULBP1 or CD48 in LAMC, including many other cancers, were significantly higher than that of the control group. It is necessary to further explain how this newly proposed P815-ULBP1+CD48 target cell inhibits the production of important NK-related active substances in blood cells isolated from MM patients.
Experiments using P815-ULBP1, P815-CD48, and P815 as controls for P815-ULBP1+CD48 should be included in all experiments.
It is necessary to further explain how this newly proposed P815-ULBP1+CD48 target cell inhibits the production of important NK-related active substances in blood cells isolated from MM patients.
NK cells isolated from MM appeared to have a significant negative effect on the expression of NK cytotoxicity in coculture with P815-ULBP1+CD48 cells than NK cells isolated from normal subjects. If the authors propose a method to overcome the negative effect of PBMC in MM patients using P815-ULBP1+CD48, it is likely to bring many readers interested in NK cell therapy into this paper.
Author Response
Reviewer 2
The authors proposed a new target cell, P815-ULBP1+CD48, as a method of measuring new NKA (NK cell activity) and tried to prove its usefulness. The NK cytotoxicity of cells isolated from PBMC or WB of normal donor (HD) and MM (multiple myeloma) patients was compared with CD107a+ cell percentage, IFN-gamma+ cell percentage, and the amount of granzyme B and IFN-gamma secreted. There was a significant difference when P815-ULBP1+CD48 was used as the target cell. When P815-ULBP1+CD48 was used as the target cell, it was more significant than when the K562 cell, which is usually used, was used as the target. The authors argue that the use of P815-ULBP1+CD48 as a target cell is advantageous for predicting treatment in patients with MM.
Analysis of LAMC cancer in the TCGA data base (by me) showed that the expression levels of ULBP1 or CD48 did not significantly affect overall survival. However, the expression level of ULBP1 or CD48 in LAMC, including many other cancers, were significantly higher than that of the control group. It is necessary to further explain how this newly proposed P815-ULBP1+CD48 target cell inhibits the production of important NK-related active substances in blood cells isolated from MM patients.
Re) In our study, ULBP1 and CD48 were selected to determine the functional capacity of NK cells, not on the basis of their role in MM progression but on the confirmed findings that their coexpression on insect S2 target cells (Y.T. Bryceson et al., Blood 2009, 114:2657, PMID: 19628705) and mouse P815 target cells (H.J. Kwon et al., Nat Commun. 2016, 7:11686, PMID: 27221592; E. Jun et al., Front Immunol. 2019, 10:1354, PMID: 31281312) synergizes for cytotoxic degranulation and IFN-g production of NK cells via NKG2D and 2B4 receptors. The rationale behind the stimulation of NK cells via NKG2D and 2B4 for assessing NK cell activity in MM was further supported by previous reports showing an association of NK cell dysfunction with a decrease in NKG2D and/or 2B4 in MM (refs 34 and 35 in the manuscript: Fauriat, C. et al., Leukemia 2006, 20:1732, PMID: 16437151; Jinushi, M. et al., Proc Natl Acad Sci USA 2008, 105:1285, PMID: 18202175). Although the role of specific ligands in MM progression merits further investigation, it is beyond the scope of our current study, which focuses on investigating the usefulness of P815-ULBP1+CD48 target cells for assessing functional capacity of NK cells in PBMC and whole blood samples. Using P815-ULBP1+CD48 target cells along with conventional K562 and 721.221 target cells, we demonstrated a significant impairment of NK cell function in MM patients, which was associated with TGF-b1 levels in the plasma rather than downregulation of NKG2D and 2B4 on NK cells in our cohort of MM patients. As suggested in previous studies (refs 36 and 37 in the manuscript: Bellone, G. et al., J Immunol 1995, 155:1066, PMID: 7636180; Viel, S. et al., Sci Signal 2016, 9:ra19, PMID: 26884601), we thus anticipate that the elevated levels of TGF-b1 might contribute to the suppression of NK cell function. However, the possible involvement of other mediator(s) or mechanism(s) in mediating NK cell dysfunction in MM cannot be excluded beyond a significant association of TGF-b1 in such context. Accordingly, we have toned down the importance of TGF-b1 in mediating NK cell dysfunction in MM and have discussed the requirement for further mechanistic study in the revised manuscript (Line 34; Line 201; Lines 220~221; Line 249; Line 303; Lines 310~312).
Experiments using P815-ULBP1, P815-CD48, and P815 as controls for P815-ULBP1+CD48 should be included in all experiments.
Re) As suggested by the Reviewer, we performed and incorporated the data analyzing the NK cell activity against wild type P815 cells, P815-ULBP1, or P815-CD48 cells as well as P815-ULBP1+CD48 cells. Consistent with previous reports, receptor coactivation with P815-ULBP1+CD48 cells resulted in synergistic degranulation and IFN-g production in NK cell population in healthy donors, whereas stimulation with wild type P815 cells and P815 cells expressing ULBP1 or CD48 alone induced little activation of NK cells (Revised Supplementary Figure 1). This data support that P815-ULBP1+CD48 target cells provide selective, uniform, and synergistic stimulation of NK cells as a useful tool to measure NK cell function. Unfortunately, PBMC and blood samples from MM patients enrolled in this study was used up and is no longer available for further functional study under our IRB approval that limited the amount of blood drawn and was expired recently. Accordingly, given limited time for revision, we have discussed the requirement for further study in a larger cohort of patients, together with the validation of P815-ULBP1+CD48 target cells in triggering synergistic activation of NK cells in healthy donors (Line 48 for an additional reference; Lines 94~95; Lines 117~118 for Revised Supplementary Figure 1).
Reference added:
- Bryceson, Y.T.; March, M.E.; Ljunggren, H.G.; Long, E.O. Synergy among receptors on resting NK cells for the activation of natural cytotoxicity and cytokine secretion. Blood 2006, 107, 159-166.
It is necessary to further explain how this newly proposed P815-ULBP1+CD48 target cell inhibits the production of important NK-related active substances in blood cells isolated from MM patients.
Re) As explained in the first response to the reviewer’s point, we anticipate the potential contribution of elevated levels of TGF-b1 to the suppression of NK cell function. However, considering the possible involvement of other mediator(s) or mechanism(s) in such context, we have toned down the importance of TGF-b1 in mediating NK cell dysfunction in MM and have discussed the requirement for further mechanistic study in the revised manuscript (Line 34; Line 201; Lines 220~221; Line 249; Line 303; Lines 310~312).
NK cells isolated from MM appeared to have a significant negative effect on the expression of NK cytotoxicity in coculture with P815-ULBP1+CD48 cells than NK cells isolated from normal subjects. If the authors propose a method to overcome the negative effect of PBMC in MM patients using P815-ULBP1+CD48, it is likely to bring many readers interested in NK cell therapy into this paper.
Re) Given a significant elevation of TGF-b1 in the plasma of MM patients along with the inverse correlation of TGF-b1 levels with NK cell function, we anticipate that TGF-b1 is a potential mediator of compromised NK cell function in MM. Although further validation is required, our findings may provide support for therapies such as therapeutic targeting of TGF-b1 aimed at restoring NK cell-mediated immune surveillance in MM. In this regard, the text has been revised accordingly (Lines 316~318).

Round 2
Reviewer 2 Report
All the questions I pointed out were answered appropriately and the results were presented. I am satisfied and agree to publish in our journal.